# Amazonian Aerosol Size Distributions in a Lognormal Phase Space: Characteristics and Trajectories

Gabriela R. Unfer[1,2,3], Luiz A. T. Machado[3,4], Paulo Artaxo[4], Marco A. Franco[4,5], Leslie A. Kremper[3], Mira L. Pöhlker[2,3,6], Ulrich Pöschl[3] and Christopher Pöhlker[3]

[1]Center for Weather Forecasting and Climate Research, National Institute for Space Research (INPE), Cachoeira Paulista, 12630-000, Brazil.
[2]Atmospheric Microphysics Department, Leibniz Institute for Tropospheric Research (TROPOS), Leipzig, 04318, Germany.
[3]Multiphase Chemistry Department, Max Planck Institute for Chemistry (MPIC), Mainz, 55128, Germany.
[4]Institute of Physics, University of São Paulo (USP), São Paulo, 05508-090, Brazil.
[5]Department of Atmospheric Sciences, Institute of Astronomy, Geophysics and Atmospheric Sciences (IAG), University of São Paulo, São Paulo, 05508-090, Brazil.
[6]Faculty of Physics and Earth Sciences, Leipzig Institute for Meteorology, Leipzig University, Leipzig, 04103, Germany.

*Correspondence to*: Luiz A. T. Machado (l.machado@mpic.de) and Christopher Pöhlker (c.pohlker@mpic.de)

**Abstract.** This study introduced a first glance of Amazonian aerosols in the $N$-$D_g$-$\sigma$ phase space. Aerosol data, measured from May 2021 to April 2022 at the Amazon Tall Tower Observatory (ATTO), were fitted by a multi-modal lognormal function and separated into three modes, the sub-50 nm, the Aitken (50-100 nm), and the accumulation mode. The fit results were then evaluated in the $N$-$D_g$-$\sigma$ phase space, which represents a three-dimensional space based on the three lognormal fit parameters. These parameters represent, for a given mode i, the number concentration ($N_i$), the median geometric diameter ($D_{g,i}$), and the geometric standard deviation ($\sigma_i$). Each state of a particle number size distribution (PNSD) is represented by a single dot in this space, while a collection of dots shows the delimitation of all PNSD states under given conditions. The connections in ensembles of data points show trajectories caused by pseudo-forces, such as precipitation regimes and vertical movement. We showed that all three modes have a preferential arrangement in this space, reflecting their intrinsic behaviors in the atmosphere. These arrangements were interpreted as volumetric figures, elucidating the boundaries of each mode. Time trajectories in seasonal and diurnal cycles revealed that fits with sub-20 nm mode are associated with rainfall events that happen in the morning and in the afternoon, but in the morning, they grow rapidly into the Aitken mode, and in the afternoon, they remain below 50 nm. Also, certain modes demonstrated well-defined curves in the space, e.g., the seasonal trajectory of the accumulation mode follows an ellipsoid, while the diurnal cycle of sub-50 nm in the dry season follows a linear trajectory. As an effect of the precipitation on the PNSDs and vice-versa, $N$ and $D_g$ were found to increase for the sub-50 nm mode and to decrease for the Aitken and accumulation modes after the precipitation peak. Afternoons with precipitation were preceded by mornings with larger particles of the accumulation mode, whose $D_g$ was ~10 nm larger than in days without precipitation. Nevertheless, this large $D_g$ in the morning seems to influence subsequent rainfall only in the dry season, while in the wet season, both $N$ and $D_g$ seem to have the same weight of influence. The observed patterns of the PNSDs in the $N$-$D_g$-$\sigma$ phase space showed to be a promising tool for the characterization of atmospheric aerosols, to contribute to our understanding of the main processes in aerosol-cloud interactions, and to open new perspectives on aerosol parameterizations and model validation.

## 1 Introduction

Representing the interactions between atmospheric aerosols and clouds is one of the major challenges in climate and Earth systems models (Forster et al., 2021). Aerosols are a key factor for cloud formation and properties (Albrecht, 1989; Koren et al., 2005; Rosenfeld et al., 2008; Heikenfeld et al., 2019), while weather events are one of the main drivers in controlling aerosol concentration (Machado et al., 2021; Khadir et al., 2023).

Extended time periods and high spatial resolution simulations are associated with significant computational costs, especially if sophisticated parametrizations of the aerosol-cloud interactions are considered. Certain processes are generalized, particularly at the expense of resolving aerosol and cloud processes (Roesler & Penner, 2010), which sometimes provide only the total particle or droplet number concentrations. By applying sensitivity tests in Amazonian warm-phase clouds, it was shown that the intensity of droplet activation is mostly influenced by a variation in diameters and standard deviations of the particle number size distribution (PNSD), rather than by the total number concentration (Hernández Pardo et al., 2019). Thus, one of the ways to better represent aerosol-cloud interactions would be to consider as many aerosol characteristics as possible in a way that computational cost is not significantly increased. A possible strategy for approaching this issue can emerge from better representations of the PNSDs themselves.

Working with PNSDs enables a statistical representation of an ensemble of particles and its spatiotemporal variability. Multiple models are useful for describing PNSDs, such as the gamma (Ulbrich, 1983) or the lognormal (Aitchison and Brown, 1957) distribution functions. There is no definite consensus on which model best describes aerosol particles, as documented by an ongoing debate (Williams, 1985; Wu and Mcfarquhar, 2018, Nyaku et al., 2020). Nevertheless, the lognormal distribution is widely used in aerosol and related communities, mainly due to its simplicity (Pöhlker et al., 2021; Boucher, 2015; Kulmala et al., 2012). The lognormal distribution is described by its moments, the geometric mean or median diameter ($D_g$), and the standard deviation ($\sigma$), describing and fitting different PNSD shapes, which includes multi-modal distributions (Heintzenberg, 1994).

The microphysical properties of an aerosol population, such as the particle fraction acting as cloud condensation nuclei (CCN), depend on particle size ($D_p$), number concentration ($N$), chemical composition, and hygroscopicity (Köhler, 1936; McFiggans et al., 2006; Reutter et al., 2009; Braga et al., 2021, Pöhlker et al., 2021). In the central Amazon and beyond, N and the PNSD are the primary factors defining the CCN concentration at a given water vapor supersaturation, whereas hygroscopicity and chemical composition are only of secondary importance (Dusek et al., 2006; Pöhlker et al., 2016; Pöhlker et al., 2018). The Amazonian submicron PNSD is characterized by a bi-modal shape during the wet season, comprising about equally strong Aitken and accumulation modes in low concentration levels, in comparison to the typical aerosol concentration in the Amazon and worldwide; in contrast to a monomodal shape during the dry season, comprising a strong accumulation and a largely covered/hidden Aitken mode, with comparatively high concentration levels (Zhou et al., 2002; Andreae et al., 2005; Pöhlker

et al., 2016; Wang et al., 2016; Varanda Rizzo et al., 2018, Franco et al., 2022). Particles between 10 and 50 nm are also detected at comparatively low concentrations, especially during the wet season, when aerosol growth events are most pronounced (Franco et al., 2022). Machado et al. (2021) showed that the PNSDs at the Amazon Tall Tower Observatory

(ATTO) are modulated by the degree of intensity of convective events, which increase the number concentration of sub-50 nm particles and decrease Aitken and accumulation mode particles, with different sensitivities when considering the two seasons and different measured heights.

Despite the strong efforts being made to understand and quantify aerosol properties, their effects have remained a major uncertainty in climate and Earth system models, which continuously motivates new studies and the development of novel

techniques (Lee et al., 2016; Seinfeld et al., 2016; Fletcher et al., 2018; Forster et al., 2021). One approach for particle characterizations not much explored yet is the evaluation of microphysical processes in a 3D phase space. McFarquhar et al. (2015) performed the first-ever study using an $N_0$-$\lambda$-$\mu$ space based on the parameters of the gamma size distribution to characterize ice hydrometeors. They observed that an ellipsoid could describe the population of the studied PNSDs, bounding the limits of the range of the parameters. The authors suggest that models could use this ellipsoid range as a parameterization

scheme since it allows multiple relationships that are easily implemented in a stochastic framework. Improving this approach, Cecchini et al. (2017) took advantage of the same phase space, however, applying trajectories on it. They studied the evolution of cloud droplet size distributions in transitioning clouds from warm-phase microphysical properties to a mixed-phase layer, revealing the effects of what they defined as "pseudo-forces". These forces represent the physical processes causing the displacements in the space. Similarly, Hernandez Pardo et al. (2021) applied the same approach to study the variability of the

spectral dispersion of droplet size distributions, proposing a parameterization for the shape parameter in bulk microphysics cloud-resolving models.

Essential information for aerosol studies in the Amazon and beyond is a robust knowledge of the total particle number concentrations and the PNSD (e.g., Guyon et al., 2003; Poschl et al., 2010; Andreae et al., 2015; Wang et al., 2016). These data are a full description of the multimodal PNSD and its variability in space (e.g., moving platforms) and time (e.g., long-

term time series). For a detailed mechanistic understanding, however, the variability of the individual modes (i.e., sub-50 nm, Aitken, accumulation, and coarse modes) relative to each other is often relevant. As the modes typically have a significant extent of overlap, multi-modal lognormal fit functions are routinely applied to display the variability of the modes separately (e.g., Hussein et al., 2005; Dal Maso et al., 2005; Kulmala et al., 2012; Pöhlker et al., 2016; Varanda Rizzo et al., 2018; Franco et al., 2022).

Here we use the multi-modal lognormal fit based on a broad statistical basis to characterize and visualize the variability of the Amazonian submicron modes across seasons, diel patterns, and meteorological events. However, we go beyond traditional approaches by utilizing a novel interpretation technique that represents the PNSDs distribution in the N-$D_g$-$\sigma$ phase space. This approach is based on the parameters of the lognormal fit, providing novel insights not only quantitatively, but also qualitatively

and temporally, and has the potential to improve current aerosol parameterizations in Earth system models. We present in this 3D space the arrangements and time trajectories of Amazonian aerosol modes in a seasonal and diurnal cycle and with the interaction with precipitation. Beyond that, we envision that future studies could also, for example, compare different aerosol populations, analyze growth events, and understand the distribution in the space under different conditions like synoptic systems, interannual variabilities (e.g. El Niño/La Niña), or even under different global warming scenarios.

## 2 Data and Methods

This study uses particle number size distributions (PNSDs) measured at the Amazon Tall Tower Observatory (ATTO) site, which is located in a pristine region of the Amazon rainforest, about 150 km to the northeast of Manaus (Andreae et al., 2015). The PNSD data were obtained by a Scanning Mobility Particle Sizer (SMPS) instrument, manufactured by TSI Inc, which is located inside a laboratory at the foot of the tower. The samples are collected at 60 m height (above the canopy) and transported through an inlet line to the SMPS on the ground. The PNSD data are then corrected to standard temperature (273.15 K) and pressure (1013.25 hPa) and for particle diffusion losses. The SMPS measurements cover a size range from 10 to 400 nm, with 104 bins, at a time resolution scan of 5 minutes. More information regarding the SMPS specifications and corrections can be found in Franco et al. (2022). The PNSD time series used in this study spans from May 2021 to April 2022, where the dry season spans from August to November, and the wet season from February to May. These definitions are based on aerosol-precipitation characterization, according to Pöhlker et al. (2016). A multi-modal lognormal fit was applied to each SMPS measurement by an algorithm specially optimized for ATTO region particles, as outlined in detail in Franco et al. (2022). This study aims to characterize the PNSDs and also evaluate their interaction with precipitation. Therefore, a dataset of rain intensity (RI, in mm.h$^{-1}$) was used. The measurements were acquired by a disdrometer (Joss-Waldvogel, model RD 50, Distromet LTDA) located at the ATTO-Campina site, nearly 4 km from the ATTO tower site. This distance between the two sites is within the typical size of thunderstorms, around 25 km in diameter (Seeley & Rompos, 2015), therefore both sites are under the same environment. Machado et al. (2021) provide a full description of the ATTO-Campina site. The instrument measures the raindrop size distribution at the surface. It can distinguish 127 classes of drop diameter, whose output is given in 20 drop size classes. The RI data are given in a 5-minute resolution.

From the RI data, we considered the maximum that occurred per day, obtaining a total of 87 days of valid data for the dry season and 86 days for the wet season. Then, PNSDs varying in hourly lags from -3 hours to +3 hours based on these maximums were derived. We also created another dataset for precipitation that occurred in the afternoon, from 13-18 Local Sidereal Time (LST), splitting into days with and without precipitation. To capture a significant number of days for both cases, a threshold of 0.5 mm.h$^{-1}$ of rain intensity was applied. A rainfall afternoon event was defined as having at least one record higher than the threshold, while a no-rainfall event had all records below the threshold. During the dry season, 55 days were considered "without precipitation" and 31 days with precipitation, while in the wet season, there were 29 and 58 days, respectively. Then,

aerosol concentration in the morning was taken based on these days, considering the hourly values from 06 to 12 LST of each day.

## 2.1 Multi-modal lognormal fit

Conveniently used to represent a particle size distribution, the multi-modal lognormal fit describes the shape of an aerosol population by using three main parameters: the aerosol number concentration $N$ (cm$^{-3}$), the mode geometric median diameter
$D_g$ (nm) and the mode geometric standard deviation $\sigma$ (nm). They are obtained by the application of the following Eq. (1) (Heintzenberg, 1994):

$$f(D_p, D_{gi}, N_i, \sigma_i) = \sum_{i=1}^{n=3} \frac{N_i}{\sqrt{2\pi}\ln(\sigma_i)} \times exp\left\{ -\frac{[\ln(D_p) - \ln(D_{gi})]^2}{2ln^2(\sigma_i)} \right\} \tag{1}$$

where $D_p$ is the particle diameter, and $i$ is the number of aerosol size modes, according to the characteristic multi-modal Amazonian particle population (Pöhlker et al., 2016; Machado et al., 2021, Franco et al., 2022).

Each SMPS scan of the time series from May 2021 to April 2022 was fitted using Eq. (1), resulting in an output of the three parameters for each mode per each time scan of 5 minutes. In central Amazon, the maximum number of modes is three, as revealed by Franco et al. (2022), thus the mode fitting algorithm was free to fit between one and three modes. The size ranges of 10 to 50 nm, 50 to 100 nm, and 100 to 400 nm are the ones that the algorithm tried to fit, which have been already shown to be representatives of the ATTO aerosol distribution (Franco et al., 2022). Thus, we use the following labels for the
Amazonian size mode distributions: sub-50 nm, Aitken, and accumulation modes. The algorithm first tried to find the modes based on the fixed ranges and then did two optimizations in order to improve the fit parameters. More details about the algorithm and the methodology are outlined in Franco et al. (2022). The separation of the Aitken mode of the literature into two, sub-50 nm and 50 to 100 nm, did not affect the behavior of our defined Aitken mode since the concentration of the sub-50 nm mode is low compared to the other modes and, at the same time, distinct features are observed when analyzing the sub-
50 nm mode by itself, as outlined by Machado et al. (2021) and also observed in this study. Therefore, the segregation of the Aitken mode into two brings rich information about the formation of new particles in the Amazon.

To avoid unrealistic fittings, only $D_g$ that matched its respective mode size range was used. In addition, only $R^2 > 0.9$ for total number concentration was considered and an outlier filter considering the top 1% (Percentile 99) was applied to all parameters. The data processing resulted in a total of 57820 useful fits, which represents 58.4% of the used data. The analyses consisting
of the seasons had around 20800 fits for the wet and around 21500 fits for the dry season. The dataset used in this study was proved to be statistically significant since it was in accordance with previous studies that used longer time series for the same region, as discussed in the Results section.

All the fit parameters found and plotted in each figure are highlighted in Table S1, where one can reproduce a PNSD curve by applying them in Eq. (1). As an example, the resulting seasonal dry and wet PNSD curves are presented in Fig. S1. In the present study, all analyses were done by computing the mean of each mode's parameters with respect to the temporal resolution used.

## 2.2 The N-$D_g$-σ phase space

The representation of PNSDs in a 3D phase space enables the visualization of a permutation of possibilities. In the N-$D_g$-σ phase space, each parameter is orthogonal to the other, as shown in Fig. 1a. In this space, each point corresponds to one fitted mode of a PNSD that can be interpreted as a curve describing the mode at a specific time. Figure 1a displays the N-$D_g$-σ space by dividing it into 8 sub-cubes with corresponding idealized curves to illustrate its interpretation. A trajectory in this space shows how the mode evolves over time. The present study considers the particle growth, the particle transport, the vertical movement, and the precipitation regimes as general pseudo-forces that modulate the displacements in the phase space. Note that the orientation of the phase space axes in the Results section was chosen to improve the visualization and highlight specific aspects of the data distribution. Therefore, the plots do not always follow the same orientation. There is also the representation of the 3D phase space in 2D as scatter plots of N versus $D_g$ with sigma as the color scale, in Fig. S2-6.

An example of a plot of the mean fit parameters representing the accumulation mode at ATTO for the dry and wet seasons is shown in Fig. 1b. It illustrates the trajectory of the seasonal back and forth between the two seasons' properties of the mode, being bridged by sub-cubes 2 and 8 of Fig. 1a. This localization means that the dry-wet trajectory presents a decrease in N and σ and an increase in the $D_g$ dimension. Since it deals with a seasonal trajectory, the precipitation regime and the horizontal transports are the main pseudo-forces causing the displacement seen. Therefore, the accumulation mode goes from a higher to a lower number concentration, from a smaller to a bigger geometric diameter, and from a broader to a narrower curve when going from the dry to the wet season. The changes in sigma and $D_g$ are of about 7.4% and 4.3%, respectively, considering the highest values as the references, which can account for significant changes in the total number concentration. All these characteristics can be seen in Fig. 1c.

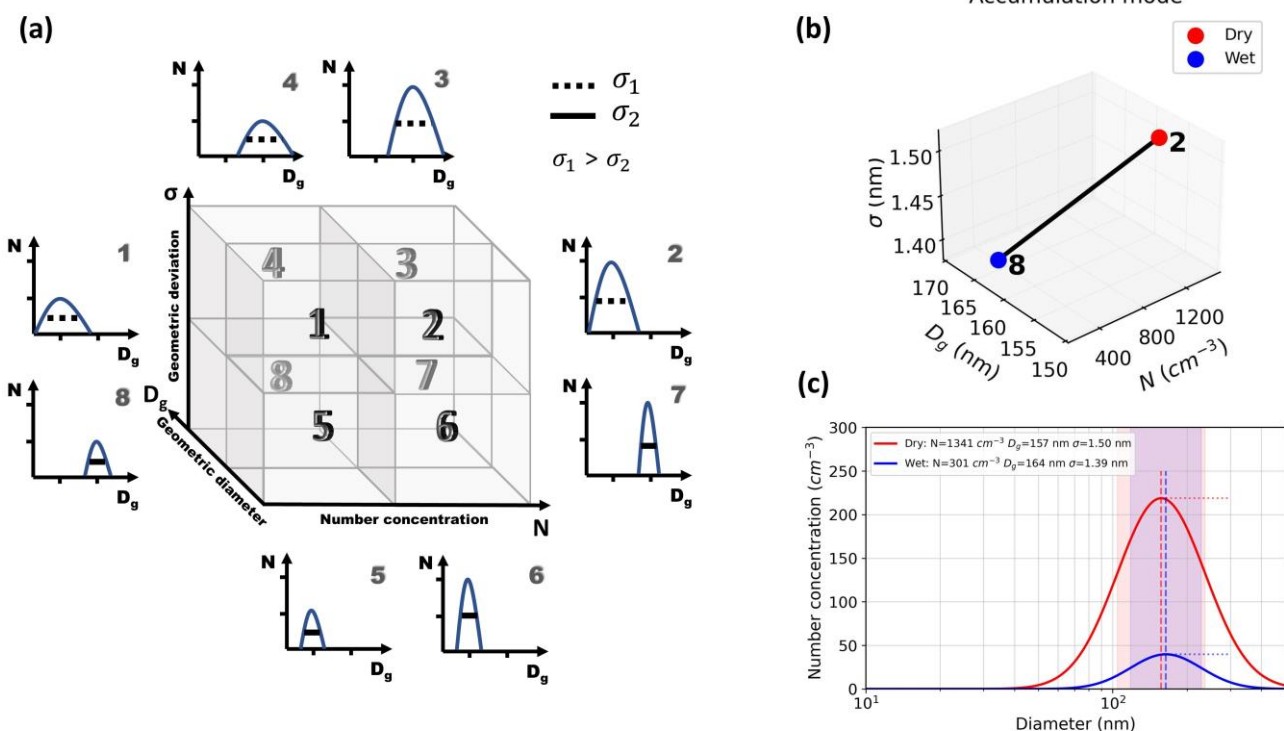

Figure 1. Conceptual drawing of the properties of the particle size distribution in the N-$D_g$-σ phase space. (a) Expected distributions when placed in specific areas of the phase space; (b) Trajectory from the dry to the wet season of the accumulation mode data used in this study. The numbers represent the location in the conceptual space; (c) The distribution of the curves using the computed mean parameters. As expected from the conceptual drawing going from sub-box 2 to 8, there is a decrease in the peak of number concentration, a displacement to a higher diameter, and a narrower curve.

## 3 Results and Discussions

### 3.1 Characterization of the PNSDs

The analysis of Amazonian PNSDs by lognormal fits with the N-$D_g$-σ phase space reveals characteristic patterns and paths in this phase space. Built on a solid basis of one year of measurements, the following results show the arrangement of all PNSDs in hourly resolution, colored by kernel density probability, where every data point represents a fitted mode.

The sub-50 nm distribution is known to have the most pronounced growth and relative concentration amplitude, especially in the wet season (Machado et al., 2021; Franco et al., 2022). By observing its configuration in the phase space (Figure 2a), one can notice a core at very small particles and very low concentration. This core comprehends PNSDs with $10 \leq D_g < 20$ nm and

$0 < N \leq 20\ \text{cm}^{-3}$, corresponding to 13.1% of the total distribution. It can indicate a starting point of the mode, which is dominated by sub-20 nm particles that can later proceed to grow into larger particles or remain as small particles. It is worth noting that the SMPS used has its lower detection limited to 10 nm, thus the complete distribution of nucleation particles, those from a few nanometers to about 10 nm, is not measured. A more profound comprehension of this core could only be done by considering the contribution of these particles, measured with specific instruments. In general, the sub-50 nm spatial distribution shows a specific pattern, characterized by a curved cone volume.

Differently from the sub-50 nm, the Aitken PNSDs show a surface with nearly constant dispersion, with a variation in $D_g$ and N (Figure 2b). They range from $D_g$ of about 70 to 90 nm, and N from 100 to 300 $\text{cm}^{-3}$, respectively, consistent with Pöhlker et al. (2016) and Varanda Rizzo et al. (2018). This variation in both $D_g$ and N can indicate that as the aerosol particles age and grow over time, which increases the dominant diameter of the distribution, there is a simultaneous change in concentration. The highest density core is centered at 200 $\text{cm}^{-3}$ with $D_g$ at 70 nm and σ at 1.2 nm. In the $ND_g$ axis surface, the density decreases radially, which shows that the central distribution is representative of the whole PNSDs, including all seasons. While on the other axes, the distribution width is more axially flattened. This arrangement characterizes a semi-sphere.

Dominated basically by N, the accumulation mode PNSDs are distributed in a geometric figure that resembles a cylinder (Figure 2c). This volume has a small range in $D_g$ and σ, and an extended range in N. This feature emphasizes the seasonal cycle of the accumulation mode, which has higher concentrations in the dry season and lower ones in the wet season, associated with smaller changes in the $D_g$ and in σ. The highest density is observed at the bottom of the cylinder, in the area of comparatively low N, and $D_g$ between 150 and 200 nm.

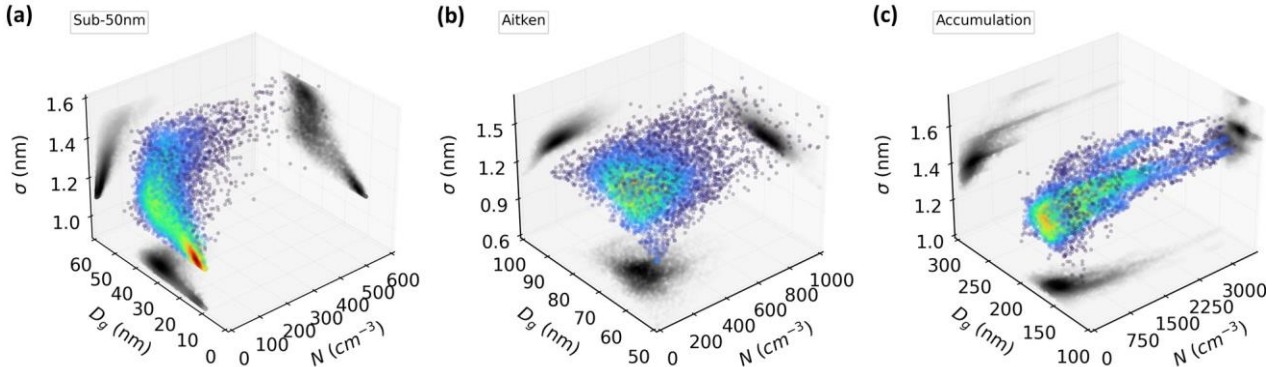

**Figure 2. Hourly Amazonian PNSDs arrangements in the N-$D_g$-σ phase space. The patterns show the distribution of the (a) sub-50 nm, (b) Aitken, and (c) accumulation modes. Every data point represents one full PNSD. In color is the density of the data points, with respective projections in grey scale shadows on all three axes. The distribution considered spans one year of measurements (May 2021 to April 2022).**

### 3.2 Seasonal trajectories

Trajectories in the $N$-$D_g$-$\sigma$ phase space allow the visualization of the displacements between PNSDs, showing the variability caused by the existing pseudo-forces. The following results show the seasonal characteristics and reveal a monthly path in the phase space. The specific distribution of each parameter per month can be seen in Fig. S7.

The accumulation mode trajectory resembles an ellipse in all three projections (Figure 3c), influenced mainly by $N$ and, to a minor extent, by $D_g$. From December-January, the seasonal cycle presents the highest $D_g$, around 175 nm, while the other

parameters are lower when compared to the whole trajectory ($\sigma$~1.4 nm, $N$~500 $cm^{-3}$). In the wet season, the trajectory is basically changed by a decrease in $D_g$, to ~164 nm. This trend goes until an inflection point in August-September, with $N$ of ~1800 $cm^{-3}$, $\sigma$ of 1.5 nm, and $D_g$ of 151 nm. This particular period marks the beginning of the dry and polluted season. The relatively high aerosol number concentration is influenced by the transition from the wet to the dry season, which starts to be characterized by a more polluted atmosphere, dominated especially by anthropogenic biomass burning (Andreae et al., 2015;

Pöhlker et al., 2018). From this point on, the trajectory returns to the initial state, with the wet season characteristics. The mean season's parameters can be found in Fig. 1c, which shows an opposite behavior regarding $N$ and $D_g$.

The dry season for the accumulation mode is characterized by high $N$, but small $D_g$, whilst the wet season presents low $N$, but bigger $D_g$. This pattern is likely due to the different sources of aerosols, from the predominance of mostly anthropogenic nearby fires in the dry season to aged long-range-transport, like African dust and Atlantic marine aerosols, in the wet season (Pöhlker

et al., 2018; Holanda et al., 2020). The trajectory revealed that the transition months, December-January and June-July, present the highest and lowest $D_g$, respectively. Also, the last two months of each season, April-May and September-October, are characterized by nearly the same $D_g$, but with different $N$ and curve width ($\sigma$). Compared with boreal forest stations in Hyytiälä and in Värriö, both in Finland and classified as pristine environments, the ATTO accumulation mode presents a smaller diameter than in these two locations, where the highest values are around 200 nm basically all over the year in Hyytiälä and

from June to November in Värriö (Tunvend et al., 2008). Even though there is a polluted season in the Amazon, the size of the accumulation particles is still comparable to pristine conditions, although the concentration is high.

The Aitken mode trajectory (Figure 3b) shows that its seasonal transition also follows the same behavior observed in the accumulation mode trajectory, presenting the extremes in $D_g$, ~74 nm (DJ), and ~67 nm (JJ). This range of diameter is basically similar to the Aitken mode found in Hyytälä (Tunvend et al., 2008). Following Pöhlker et al. (2018), during biomass burning

in the dry season, the accumulation mode dominates over the Aitken mode regarding $N$, while during the wet season, which has a near pristine environment, the Aitken mode is more equivalent (also seen on Fig S1). From Fig. 3b, the effects of this oscillation are perceived. The wet season ($N$~280 $cm^{-3}$, $\sigma$~1.31 nm, $D_g$~69 nm) is characterized by an approximately constant $D_g$ and $\sigma$, a function of basically $N$. On the other hand, the dry season ($N$~300 $cm^{-3}$, $\sigma$~1.30 nm, $D_g$~70 nm) is characterized by nearly constant $N$, increasing in $D_g$. This behavior can elucidate the circular pattern seen in the $ND_g$ axis surface in Fig. 2b.

The sub-50 nm mode monthly trajectory (Figure 3a) is limited to a $D_g$ range between ~26 and ~32 nm, nearly the same gap between the two density cores seen in the $ND_g$ axis surface in Fig. 2a. Although the core of smaller PNSDs (sub-20 nm) contributes to only 13.1%, the two points in Fig. 3a, representing the wet season, FM and AM, are the ones with a shift to smaller $D_g$. It is indeed during the wet season that most of these sub-20 nm modes are happening, as seen in Fig. S8a, representing about 20% of all PNSDs. New particle formation has been reported to happen mostly in this season, associated

with rainfall events (Wang et al., 2016; Andreae et al., 2018; Franco et al., 2022). Therefore, the core seen can be related to distributions capturing the formation of these particles.

A notable behavior is that sub-50 nm PNSDs in both wet and dry seasons show the same N, changing the $D_g$ of the distribution. The dry season is characterized by $D_g$ ~30 nm, while the wet season has $D_g$ ~27 nm. The sub-50 nm PNSDs seem to be most affected by the transition regimes, presenting the two extremes in N. The peak observed in JJ, the period of clean to polluted

transition, is shifted to larger particles, where a major part of the second core in larger particles and higher concentration observed in Figure 2a, could be associated with this period.

By analyzing the three modes' trajectories, all of them presented extremes either in N or $D_g$ in transition months. This feature is likely related to the effect of rainfall regimes on size distribution, since sub-50 nm increases its concentration, and Aitken and accumulation modes decrease during rainfall events (Machado et al., 2021).

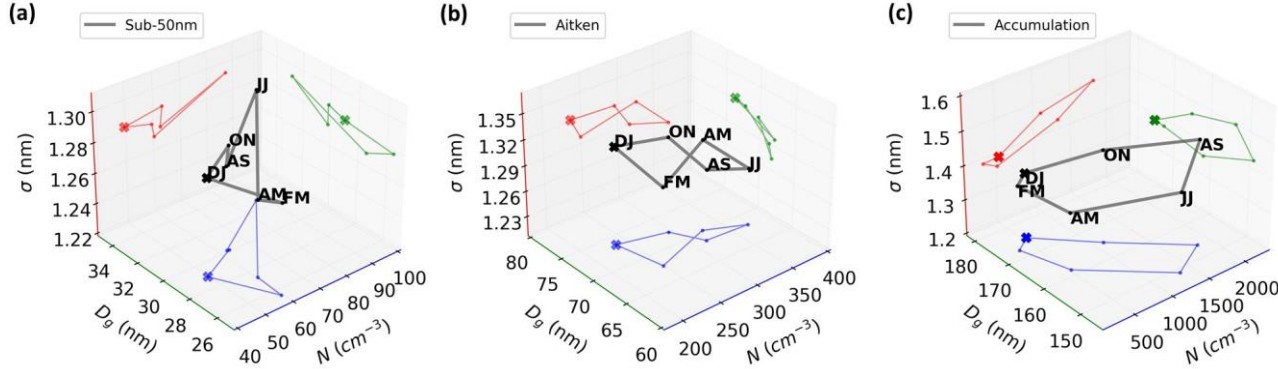


**Figure 3. Seasonal trajectories of Amazonian PNSDs in the N-$D_g$-σ phase space. The trajectory is shown in bimonthly resolution, showing the patterns of the (a) sub-50 nm, (b) Aitken, and (c) accumulation modes. The cross symbol marks a visual reference and the colored trajectories are projections in each axis surface.**

### 3.3 Diurnal cycle trajectories

Machado et al. (2021) showed that the ATTO diurnal cycle of the number concentration of the three aerosol modes is well-defined. The authors revealed that sub-50 nm mode N is strongly controlled by the diurnal cycle, reaching a maximum during the night probably due to late afternoon rainfalls and the establishment of the nocturnal boundary layer that keeps the concentration nearly constant, and reaching a minimum as the sun rises when particle growth initiates, which is also

corroborated and described by Franco et al. (2022). Aitken and accumulation modes follow an opposite behavior, increasing in N with the sunrise, reaching a maximum in nearly the middle of the afternoon, starting to decrease due to dry deposition and scavenging, until achieving a minimum during the night and early morning. While in the SMEAR II station in Hyytiälä, for the period of November to January (dry season in the Amazon), Mäkelä et al. (2000) showed different modes are stable in size during the diurnal cycle, and from February to May (wet season in the Amazon), growth between the modes was observed, especially from the nucleation to Aitken mode. The following results (Figure 4) show the ATTO modal diurnal cycle in the wet and dry seasons and unfold the link to diameter change. To show the trajectories more clearly, the times were plotted by every 6 LST.

The sub-50 nm mode diurnal cycle in the dry season (Figure 4a) shows a strong link to all three parameters. The trajectory changes linearly from midnight to noon, and vice-versa. The concentration decreases until the minimum at 12 LST and increases until the maximum at 00 LST. The $D_g$ and $\sigma$ also follow this pattern, with a minimum $D_g$ of about 28 nm and a maximum of about 32 nm. When the sun rises (06 LST), particle growth starts to dominate, given that N and $D_g$ decrease. The primary distinction between the dry and the wet seasons lies in the range of $D_g$ observed. Specifically, during the wet season, the diameter amplitude tends to be lower, which can be attributable to the prevalence of sub-50 nm rapidly growing into Aitken particles and/or by distributions without growth. Conversely, during the dry season, the advection of particles may contribute to the greater variability in the diameter of the particle size distribution. The second core seen in larger sub-50 nm PNSDs (Figure 2a), could be partly associated with diurnal features of the dry season.

Franco et al. (2022) pointed out a value of 88% of growth events of sub-50 nm associated with new particle formation happening in the wet season. The diurnal trend specifically for these cases showed a peak in concentration in the early morning, which could explain the peak seen in Fig. 4d at 06 LST. Comparing with Fig. 2a, where a core around 20 nm is observed, the diurnal cycle does not show average diameters around this value. This could be associated either with the low frequency of occurrence of these sub-20 nm PNSDs (only 13.1%) or because of being equally distributed throughout the day. Figure S8b elucidates this fact by revealing that there is actually a diurnal pattern, but indeed happening at a low frequency. This diurnal cycle reinforces that these sub-20 nm PNSDs are produced by rainfall events since the pattern seen resembles a rainfall diurnal cycle, that mainly happens in the afternoon, but also shows a peak in the morning. This feature is particularly different from new particle formation events in the boreal forest region of Hyytiälä, Finland, where nucleation events are majority associated with non-cloudy conditions (Dada et al., 2017; Sogacheva et al., 2008).

The diurnal cycle for the Aitken mode is different among the seasons. In the wet season (Figure 4e), the whole cycle occurs at nearly constant N, with a change in the $D_g$ and $\sigma$, that decreases from 00 to 12 LST with a following increase from 12 to 18 LST. The concentration stability could be explained by a balance between the aerosol growth events from the sub-50 nm to the Aitken mode and from the Aitken to the accumulation. In contrast, the changes in diameter are dominated first by growth events from the sub-50 nm to the Aitken mode in the morning (06-12 LST) and second by growth from the Aitken to the

accumulation mode in the afternoon (12-18 LST). This behavior coincides with the decrease in N and the increase in $D_g$ from 06-12 LST in the sub-50 nm cycle (Figure 4d), and with the $D_g$ growth in the accumulation mode cycle (Figure 4f) from 12-18 LST. In other words, in the morning, it seems to have no other source of Aitken particles than the growing process from sub-50 nm. In the afternoon, Varanda Rizzo et al. (2018) showed that there is a source of Aitken mode associated with the transport by downdrafts into the boundary layer, although it can also act as a sink due to the association with precipitation. This source from downdrafts could explain the stability in N during the afternoon.

These findings conclude the comprehension of the core in sub-20 nm (Figure 2a): a substantial contribution for the density is of a prevalence of sub-20 nm PNSDs that remain with small particles, occurring in the afternoon, and a minor contribution is of sub-20 nm PNSDs that rapidly grow into Aitken, happening in the morning.

For the dry season, the Aitken mode diurnal cycle evolution (Figure 4b) is different from the wet season, and the main variation is at the particle number concentration (N). During the night and early morning, the particles are confined inside the nocturnal boundary layer and reach the maximum concentration. During the day, with the development of the boundary layer, the aerosol number concentration is reduced, and their size increases, also showing a growing process to accumulation mode. Unlike the wet season, in the dry season, the growth rate from Aitken to accumulation mode seems to be faster and the most predominant process rather than the growth from sub-50 nm to Aitken, resulting in a decrease in the Aitken number concentration during the growing process. This is in accordance with Franco et al. (2022), who showed that only 12% of sub-50 nm growth events happen during the dry season.

The accumulation mode in the dry and wet seasons (Figures 4c and 4f) have nearly the same pattern, only shifting in N and in $D_g$ according to the season's features. Note that the axes are rotated, and disposed differently from the others to improve the visualization. The cycle is characterized by a minimum in both N and $D_g$ at 06 LST, followed by an increase until 12 LST, corresponding to the time when the Aitken mode is reduced, and a following decrease until back to sunrise (06 LST), probably associated with the afternoon rainfall and the scavenging processes. The $D_g$ variation goes from around 154 nm to 158 nm in the dry, and from about 162 nm to 165 nm in the wet season. The minimum and maximum concentration match with the sink and growth process of the day (06 and 12 LST), respectively.

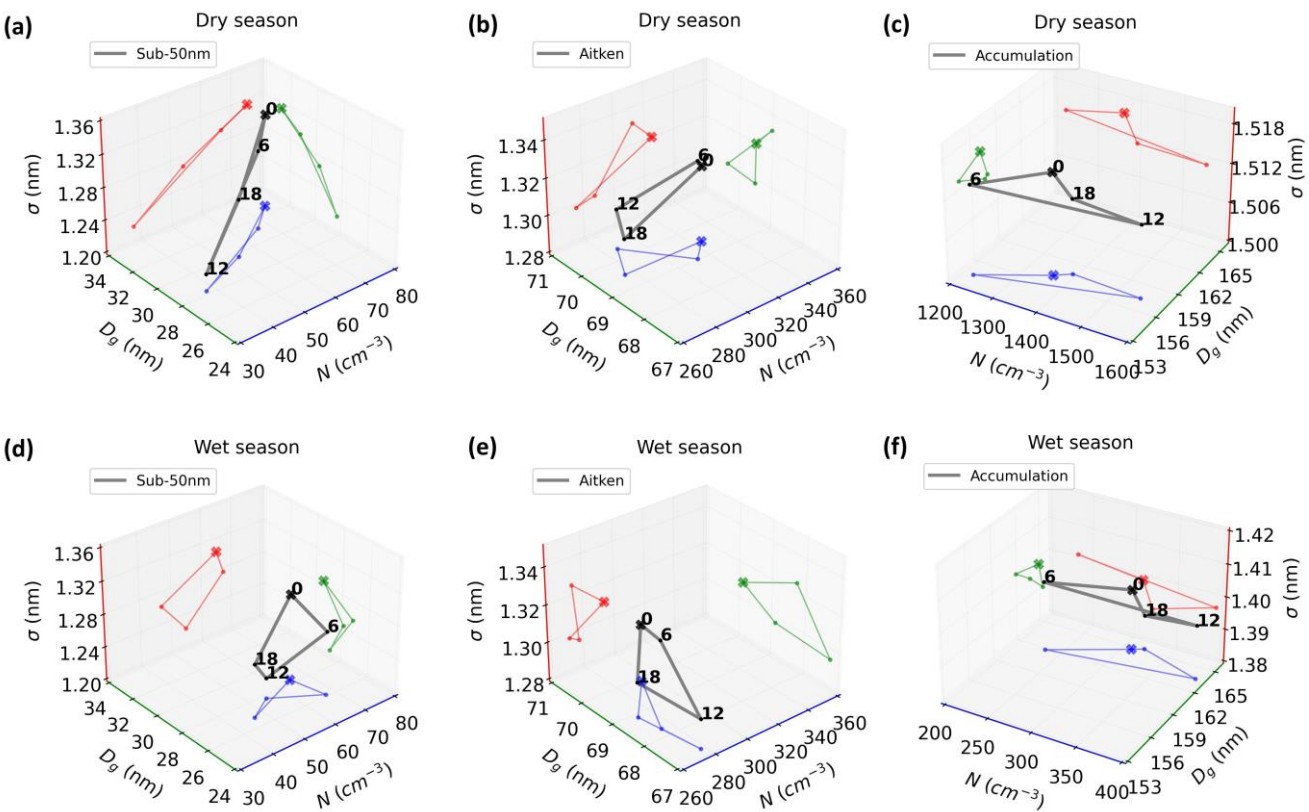


**Figure 4. Diurnal cycle trajectories of Amazonian PNSDs in the N-D$_g$-σ phase. The trajectories show the patterns of each mode (columns) for each season (rows). The numbers in black represent the time in LST, the cross symbol marks a visual reference, and the colored trajectories are projections on each axis surface. The cubes (c) and (f) are slightly turned to better visualize the trajectories.**

**3.4 Effects of precipitation on PNSD**

Machado et al. (2021) studied the relationship between lightning and PNSD and found out that the Aitken and the accumulation modes concentration decrease at the same time sub-50 nm mode increases with the maximum intensity of lightning. This behavior begins to occur 100 minutes before the maximum lightning density, indicating this moment as the beginning of the rainfalls, which match its maximum with maximum lightning density. Wang et al. (2016) also observed a direct relationship

between convective events and ultrafine aerosol particles in the central Amazon. Both studies characterize the course of the particles regarding their number concentration. The following discussion is based on precipitation maximums, which can be intercomparable to the findings of Machado et al. (2021), since the peak of maximum lightning and precipitation occurs simultaneously (Mattos et al., 2017), promoting a complete characterization of the aerosol-precipitation interaction. Note that for better visualization, the accumulation mode axes are rotated (the same configuration that was presented in Figures 4c and

4f).

In the dry season, sub-50 nm (Figure 5a) N keeps constant at around 60 cm$^{-3}$ increasing only the dominant $D_g$ also at constant $\sigma$, until maximum precipitation. From the precipitation peak on, the PNSDs start to increase in $D_g$ after 1-2 hours and reach the highest N 3 hours later. In the wet season (Figure 5d), the post-maximum increase in N and $D_g$ is also seen, although the maximum occurs 2 hours later. It is at 2 hours before and after that the two extremes in $D_g$ occur, reaching the minimum and maximum, respectively. This might be related to the start and end of the precipitation event, nearly the same moments seen by Machado et al. (2021). This feature is more pronounced in the wet than in the dry season.

Many studies have reported that in the wet season, sub-50 nm particles increase in concentration after a precipitation event (Wang et al., 2016; Andreae et al., 2018; Franco et al., 2022). The results here confirm this increase, also observing the same behavior in the dry season, although with a lower increase. In addition to the concentration increase, Figures 5a and 5b also show that the mode itself increases in size. Therefore, there is a parallel increase of the diameter and the concentration of the sub-50 nm mode after precipitation events.

Larger particles are washed out by precipitation, therefore, a decrease in concentration is expected. This behavior is especially pronounced in the wet season, where higher amounts of precipitation are observed (Machado et al., 2004). In fact, our results display this decrease for the Aitken and the accumulation modes. Figures 5e and 5f show a decrease in the concentration after the precipitation peak, seen from the circle to the square markers, with a higher decrease in the bigger particles, the accumulation mode. However, the results revealed another feature, where the modes are also characterized by bigger and smaller particles, before and after the maximum, respectively.

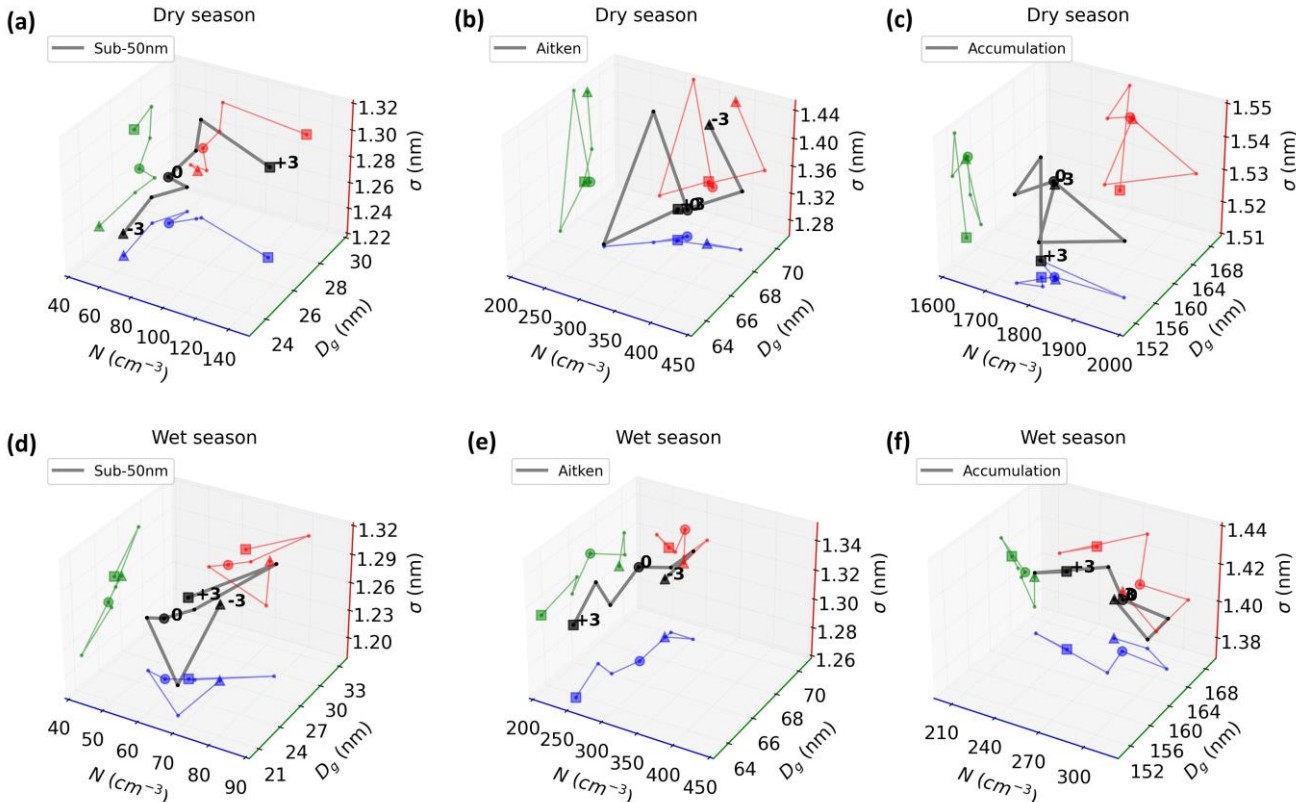

**Figure 5. Composites of Amazonian PNSDs in the N-D_g-σ phase space for before and after maximum precipitation. The trajectories span from 3 hours before (triangular symbol) to 3 hours later (square symbol), considering the moment of maximum precipitation (circle symbol), for each mode (columns) and each season (rows).**

### 3.5 Trends in the accumulation mode PNSDs trajectories when associated with precipitation occurrence

The following analysis explored the background aerosol concentration in the morning considering afternoons with and without precipitation. It was considered time trajectories from 6 to 12 LST. The aerosol mode that has the most influence on precipitation, acting as CCN, is the accumulation mode. Initially, the analysis was performed for all three modes, however, the sub-50 nm mode did not exhibit any difference in the early morning trajectories (not shown). The Aitken mode slightly displayed some differences (not shown), but effectively, the accumulation mode was the one that clearly showed different patterns. Therefore, the following analysis of Figure 6 was done exclusively for this mode, which exhibited clear preferential characteristics and trajectories in the phase space.

In the dry season (Figure 6a), the early morning trajectories of both cases show an increase in concentration, characterized by the diurnal cycle, at different rates though. Days with precipitation in the afternoon are preceded by noons with lower particle number concentrations. In the dry season, the Amazonian PNSD is monomodal, dominated by the accumulation mode with a large concentration (Figure S1). Nevertheless, aerosol concentration is so high that the amount of these particles needed for

CCN activation seems to no longer be a causal factor. In contrast, bigger particles, which increase the dominant $D_g$ of the
PNSDs, seem to be present in the mornings with precipitation in the afternoon. Therefore, in the dry season, the diameter of
the aerosols appears to have the most influence on precipitation to occur, rather than the concentration.

For the wet season (Figure 6b), the accumulation mode shows the same general pattern as in the dry season. It presents the
same concentration trajectory for both cases, with a shift to larger particles in rainfall cases, but with a concentration trajectory
that goes further. This shows that due to the lower number concentration in the wet season, both concentration and diameter
have an equal influence on precipitation. Machado et. al (2021) showed that in the wet season, a background of high
accumulation mode concentration is associated with a maximum activity of lightning events 150 min later, possibly indicating
an invigoration of convective clouds. Based on Fig. 6b, it seems that this high-concentration background is also ruled by larger
accumulation particles.

In general, rainy days have shown a $D_g$ in the morning increased by ~10 nm compared to days without precipitation. This same
feature is also present in Fig. 5, confirming that precipitation events are preceded by larger accumulation particles. Hernández
Pardo et al. (2021) showed that Amazon clouds in a more polluted environment presented a smaller variability in droplet size
distribution width due to CCN being already large enough to produce droplets. This large size seen in the morning seems to
be associated with the factors that can promote precipitation in the afternoon, of course probably along with other favorable
conditions, such as high humidity. The observed features highlight the importance in considering in aerosol parameterization
not only the total particle concentration, but also the aerosol-size distribution.

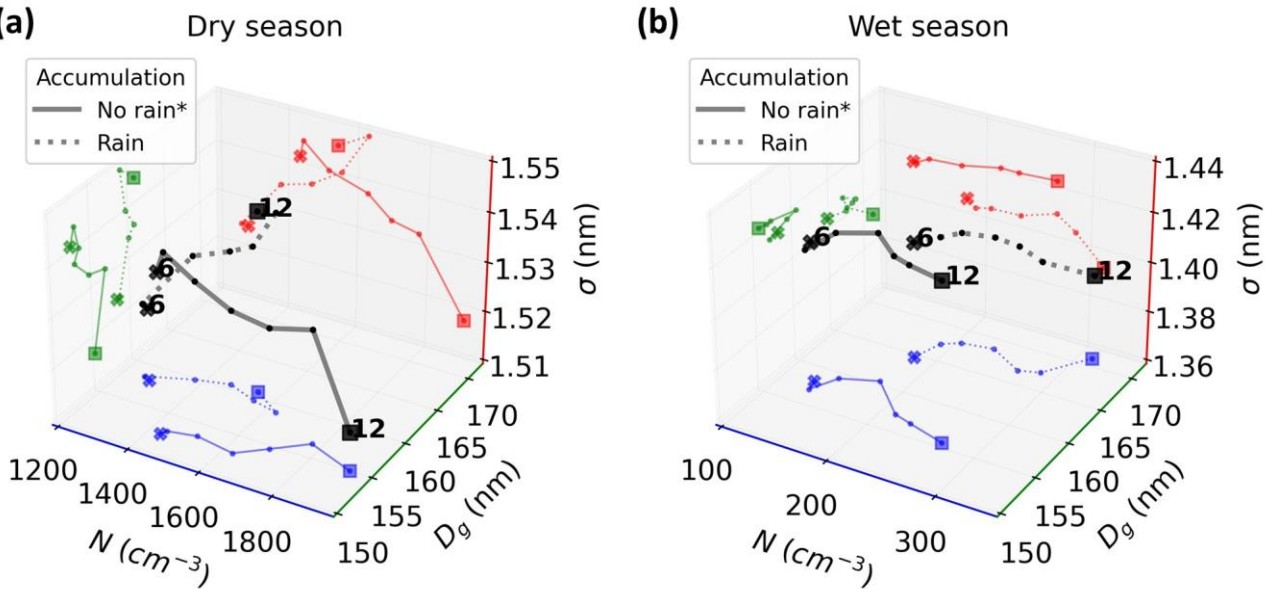

**Figure 6. Composites of Amazonian PNSDs in the N-$D_g$-σ phase space for days with and without precipitation. The trajectories are in the morning, going from 6 (cross marker) to 12 (square marker) LST, for cases of afternoons with and without rainfall events.**

 **An afternoon with rainfall was defined as having at least one record of rain intensity (RI) ≥ 0.5 mm.h$^{-1}$ from 13 to 18 LST, while a no rainfall afternoon was defined as having all records of RI < 0.5 mm.h$^{-1}$.**

## 4 Summary and conclusions

Amazonian particle number size distributions (PNSDs) were analyzed by implementing a new approach, the N-D$_g$-σ phase space, a 3D space based on the fit parameters of a multi-modal lognormal function. The parameters are the number concentration (N), the geometric median diameter (D$_g$), and the geometric deviation (σ). The fit was applied to in situ SMPS data covering 1 year, measured at the ATTO tower, in central Amazonia. Visualizing the arrangements of each mode in the phase space enabled a volumetric characterization of a population of PNSDs and a time evolution analysis by trajectories.

In the phase space, the sub-50 nm PNSD population was represented by a curved cone volume, the Aitken PNSDs by a semi-sphere, and the accumulation PNSDs by a cylinder. This is the first glance of the volumetric characterization of Amazonian aerosol mode distributions. The knowledge of such volumes bounds the limits of these distributions, elucidating all possible solutions of a PNSD, providing the tools for parameterization development.

The seasonal trajectory of the accumulation mode PNSDs in the N-D$_g$-σ has a well-defined cycle, characterized by an ellipsoid. The extremes in D$_g$ values occur during the transition months, December-January (172 nm) and June-July (146 nm), presenting the highest and lowest values, respectively. The highest value is around thirty nanometers smaller than what is seen throughout the year in the pristine boreal forest in Hyytälä. This means that even having a polluted season, the ATTO site has accumulation mode particles comparable in size to pristine environments. The fit parameters for the dry season were N=1368 cm$^{-3}$, D$_g$=157 nm, and σ=1.50 nm, while for the wet season N=285 cm$^{-3}$, D$_g$=164 nm, and σ=1.39 nm. The Aitken mode PNSDs in the wet season are characterized by an approximately constant diameter (D$_g$~69 nm) and deviation (σ~1.31 nm), being a function of basically N, while in the dry season, it is characterized by nearly constant concentrations (N~300 cm$^{-3}$), increasing in diameter. The range of the Aitken mode diameter is basically analogous to what is observed in Hyytälä. The seasonal trajectory of the sub-50 nm mode PNSDs presented a similar number concentration (N=60 cm$^{-3}$) in both dry and wet seasons, changing the diameter, D$_g$~30 nm, and D$_g$~27 nm, respectively.

The diurnal cycle during the dry season of the PNSDs of sub-50 nm mode follows a linear cycle in all three axes with respect to time. The trajectory decreases (increases) in all three parameters from 00 to 12 LST (12 to 00 LST). In the wet season, the cycle shrinks, and there is less variation in the D$_g$. The Aitken mode PNSDs diurnal cycle has different patterns when comparing the two seasons, characterized by a nearly constant N in the wet season. In contrast, in the dry season, it is the diameter that remains constant. This behavior in the wet season can be explained by an equilibrium in concentration gain and loss due to the growing process from the sub-50 nm to the Aitken mode and from the Aitken to the accumulation mode. Regarding the changes in diameter, the first process dominates in the morning and the second in the afternoon. This feature indicates no other source of Aitken mode particles in the morning than the growing process from sub-50 nm. In the dry season,

the growth from the Aitken to the accumulation mode dominates, promoting a decrease in N. For the accumulation mode PNSDs, the diurnal cycle seems to depend on N, with few variations in $D_g$ and $\sigma$, in both seasons. The cycle's extremes in N are at 06 LST (minimum) and 12 LST (maximum), matching with sink and growth processes that occur during the day.

A density core with sub-20 nm modes was observed in the arrangements of all sub-50 nm PNSDs, representing 13.1%. This core was associated with a substantial occurrence of these PNSDs during the wet season, especially from PNSDs remaining

as sub-20 nm, happening during the afternoon. A minor occurrence of these sub-20 nm PNSDs was associated with a rapidly growing process to the Aitken mode, which happens in the morning. These occurrences of the sub-20 nm distributions in the morning and in the afternoon could be linked to the occurrence of precipitation since, in central Amazon, the precipitation follows the same pattern and is known to be associated with new particle formation. Future studies should focus on investigating sub-10 nm particles and their contribution to the particle growing process.

Finally, the two-way effect of precipitation and PNSDs was explored. After maximum rain intensity, sub-50 nm PNSDs not only increase in N but also in $D_g$. This increase starts after 1 hour, going up to 2 hours after the rain peak, suggesting that this effect is due to new particle formation or sub-10 nm particle growth. At the beginning of the precipitation events, sub-50 nm PNSDs reach a minimum in $D_g$. For Aitken and accumulation mode, a decrease in N and $D_g$ was seen after maximum precipitation, and before the maximum, N and $D_g$ were higher. By analyzing how accumulation mode PNSDs background

trajectories in the morning would affect the precipitation during the afternoon, we concluded that how big the accumulation particles are can be associated with the possibility of precipitation to occur. Days with afternoon precipitation had early morning accumulation mode particles larger than days without precipitation. The $D_g$ is increased by ~10 nm. The trajectory revealed that for both cases there is an increase in N from 06 to 12 LST, however, it is only in the wet season that both N and $D_g$ seem to be one of the factors for precipitation to occur. While in the dry season, it is exclusively the $D_g$, as the particle

concentration background in the dry season is already very high, larger than 1000 cm$^{-3}$. Further investigations must be done linking this diameter effect with thermodynamical conditions.

This study provided an overview of Amazonian PNSDs variability in the N-$D_g$-$\sigma$ phase space and its trajectories, which facilitated the physical interpretation of processes involving particle source, sink, and growth, and elucidated possible aerosol-cloud parameterizations. Future studies could make use of the same approach, for example, to compare different aerosol

populations, analyze multiple growth events, evaluate clusters, extract parameterizations, and validate Earth systems models.

**Data availability**

The data that support the findings of this study are openly available in Edmond repository at
https://doi.org/10.17617/3.4DBT3W.

## Author contributions

GRU and LATM designed the study. GRU processed the data. GRU, LATM, and CP wrote the manuscript. MAF, PA, UP, LK, and MLP provided valuable ideas for the data analysis and the interpretation of the results. LATM and CP supervised the study. All authors revised the manuscript.

## Competing interests.

The authors declare that they have no conflict of interest.

## Acknowledgments.

We would like to thank the financial support done by the São Paulo Research Foundation (FAPESP), by the Coordenação de Aperfeiçoamento de Pessoal de Nível Superior - Brasil (CAPES), by the Bundesministerium für Bildung und Forschung, by the Brazilian Ministério da Ciência, Tecnologia e Inovação, and by the Max Planck Society. For the operation of the ATTO site, we acknowledge the support of the Instituto Nacional de Pesquisas da Amazônia (INPA), the Amazon State University (UEA), the Large-Scale Biosphere-Atmosphere Experiment (LBA), FAPEAM, the Reserva de Desenvolvimento Sustentável do Uatumã (SDS/CEUC/RDS-Uatumã), and the Max Planck Society. Particularly, we would like to thank the ATTO-Campina team involved in the technical, logistical, and scientific support. We would like to thank Daniele Visioni and the anonymous referee for the comments and recommendations that improved the quality of the paper.

## Financial support

This research has been supported by the São Paulo Research Foundation (FAPESP), grants #2021/03547-7, #2022/01780-9, and #2022/07974-0, by the Coordenação de Aperfeiçoamento de Pessoal de Nível Superior - Brasil (CAPES) - Finance Code 001, by the Bundesministerium für Bildung und Forschung (BMBF contracts 01LB1001A, 01LK1602B, and 01LK2101B), by the Brazilian Ministério da Ciência, Tecnologia e Inovação (MCTI/FINEP contract 01.11.01248.00) and by the Max Planck Society.

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
