# Peer review of "Amazonian Aerosol Size Distributions in a Lognormal Phase Space: Characteristics and Trajectories"

_EGUsphere, 2023_

## Author Comment (AC1)

**Response to Daniele Visioni**

**Article: "Amazonian Aerosol Size Distributions in a Lognormal Phase Space: Characteristics and Trajectories", by Gabriela R. Unfer et al., Egusphere 2023-1361.**

Dear Editor,

The authors thank both reviewers for their helpful comments and suggestions. Our responses to each comment are developed hereafter, along with an indication of changes made in the revised version of the text. As a summary, the revisions to the manuscript include the following highlights:

- All plots of the Results section have a version in 2D as scatter plots of N versus $D_g$ with sigma in a color scale presented in the Supplement.

- A table containing the summary of all fit parameters is presented in the Supplement.

- A more detailed description of the code-fitting methodology and a discussion about the separated analysis of the modes sub-50 nm and 50-100 nm (Aitken).

- The text was improved in readability and the abstract was rewritten.

- New references were added to improve the discussions.

 The individual reviewer comments and responses are included in the following document, where reviewer comments are presented in **bold** and the author comments in *italics*.

Sincerely,

Gabriela R. Unfer, on behalf of all co-authors

**Reviewer 1: Daniele Visioni**

**General:**

**This study illustrates a new method to visualize particle number size distributions for atmospheric aerosols, and presents results from one year of data collected at the Amazon Tall Tower Observatory. The authors did a good job with the framing (very neat and clear introduction), and while there are places where the writing could be slightly improved (see below for some suggestions), the manuscript is pretty clear and undeniably useful.**

*Dear reviewer, we would like to thank your comments and suggestions; they were significant for improving and clarifying essential aspects of the manuscript. In the Editor's letter, we explain the main changes in the manuscript, and below, we listed the aspects related to your recommendations.*

**Three major comments: 1) 3D pictures are hard to understand and visualize, and 2D projections could be done better here. Descriptions of the 3D shape are sometimes puzzling/unscientific. I commented more on this below.**

*We considered your suggestion carefully and made 2D plots for some of the original plots, as shown below. We first obtained a scatter plot of N as a function of $D_g$ with sigma in a color scale. In addition, we tried another kind of plot, a ternary one, to check its implementation.*

*For the scatter plot, the valuable information that the 3D plot shows was mainly lost: the identification of different aerosol particle arrangements along with the space phase. The idea of perceiving the delimitation in this space is to offer a first glance for model validation. Moreover, knowing that a mode distribution follows a certain pattern is a relevant information, especially when future studies compare different aerosol populations for different locations of the globe. On the other hand, the 2D plots are simply the projections plotted in the 3D as colored lines. The main idea of making the projections was to improve the visualization, where one can see each projection and look into the values in detail. At the same time, the plot itself gives the pattern information. However, to provide a second option for the reader to analyze the phase space results, we added in the supplement the 2D plots (Figures S2-6).*

*We made a ternary plot where the variables are normalized in the way that they all vary from 0 to 1 and each axis is one of the three variables we studied (N, $D_g$, and sigma). This new plot compromised the interpretation. The information we derived from the 3D plots is not easily seen, although it might be an interesting tool for future analyses. For our present study, though, we do not think it is applicable.*

[Figure]

**Figure 2 - Density**

Colors: kde density

Normalization
(X-X.min) / (X.max − X.min)

Colors: kde density

[Figure]

**Figure 3 - Seasonal**

[Figure]

**2) There is a lack of analyses in terms of statistics, so hard to tell how significant some results are. A problem in the figures arising from the 3D representation, but both the 2D projections could do with some measure of significance, and mainly the numbers reported could be better framed. I offer an example in my comment below for Figure 1.**

*We believe that the 3D plot is sufficient to describe our results, which is why we have it in the manuscript, but we are presenting the 2D in the Supplement (Figures S2-6). In order to make the statistics clearer, we added the actual values of each plot as a table in the Supplement (Table S1), which improves the interpretation of the results and evaluates qualitatively the differences among the modes and case studies.*

**Finally, 3) while I want to remark that the manuscript is mostly well written and the results could be of interest so this should not be an obstacle itself to publication, I am unsure if it fits into the category of a "Research Article" or if this should more neatly fit into a "Measurement reports" which ACP clearly states "Analysis of the measurements may include model results and conclusions of more limited scope than in research articles." This seems to me to be fitting here: it's only 1 year of measurements, and the statistics are unclear (maybe the authors can convince me otherwise if they present them). Furthermore, while the abstract starts by highlighting the novelty of the approach, the conclusions mostly highlight the results. I think this is up to the Editor to decide, but I would urge the authors to consider changing the scope to a Measurement report so that they can more clearly state what is the relevant part of the manuscript (the approach), offer the year of measurements as a result stemming from the approach, but then more clearly state that future work is needed to produce more statistically robust results. Whatever the decision over the manuscript type, which is not entirely my business to decide, I suggest an eventual publication in ACP after the text and figures are made clearer and after some statistical analyses have been performed.**

*Our results are very robust and reproduce the main particle size distribution features, certifying the data quality and our research. Our results agree with studies that used longer time series of the central Amazon (e.g. Franco et al., 2022, Pöhlker et al., 2016, Varanda Rizzo*

*et al., 2018). The study of Franco et al. (2022) used a time series from February 2014 to September 2020, more than six years of data, and yet our analyses are intercomparable and in agreement with each other. In addition, the most recent data were also collected with more sophisticated instruments and preprocessing techniques, as adjusted for standard temperature, correction due to inlet losses, and frequent instrument calibration.*

*Our study's idea is to present the approach of the lognormal phase space and also analyze and report the new findings. This is the first study to use this approach for analyzing Amazonian atmospheric aerosols, where we could obtain new knowledge about particle size distribution variability and show a different perspective from what was already known. We highlight the following new results:*

*1) The lognormal phase space is a promising tool for analyzing aerosol distributions, including comparing different aerosol populations and stages of aging.*

*2) Aerosol modes have a preferential arrangement in this space, reflecting their intrinsic behaviors in the atmosphere.*

*3) Amazonian new particle formation, showing a core in sub-20 nm fits, is associated with morning and afternoon rainfall events, following the same diurnal precipitation cycle. While the morning sub-20 nm fits rapidly grow into the Aitken mode (50-100 nm), the afternoon distributions remain below 50 nm.*

*4) In the wet season, the source of the Aitken mode particles in the morning is the growth from the sub-50 nm particles, while in the dry season, the source is from the confined particles of the previous nocturnal boundary layer.*

*5) After rainfalls, the sub-50 nm mode distribution increases not only in concentration but also in diameter. Comparing before and after the precipitation, Aitken and accumulation particles are bigger and in higher concentration before the precipitation, and smaller and lower, after.*

*6) When considering the effects of the aerosol number concentration and the diameter of the accumulation mode particles, both parameters are likely to have a similar influence on precipitation occurrence in the wet season. In contrast, in the dry season, only the diameter likely has an effect. It was observed that accumulation mode distributions had bigger geometric diameters on the morning of rainy afternoon days than on days without precipitation. The geometric diameter was larger by ~10 nm in both seasons.*

*Therefore, we truly believe that our manuscript fits perfectly into the "Research Article" category since we present results and conclusions of significant impact for the Amazon rainforest and the whole aerosol community.*

**Specific comments:**

**85: "data" is here first used as singular and then as plural. I have no strong opinions over the debate, but consistency is probably better either way!**

*This referred word was changed by the word "information" and we checked the rest of the text to ensure it is grammatically consistent.*

**135: 57820 out of how many?**

*The total data is composed of 98939 fits obtained on a time step of 5 minutes. Then, after the filters described in the manuscript were applied to the data, there were 57820 valid data, representing 58.4% of the used data. This information was added to the text.*

**Fig. 1: The figure could be improved for clarity. The 3D numbers in the cube are not really visible. Panel a) does not have the sigma_2 line very visible (and not sure the color matches). Sigmas are also not very clear in Panel c). Lastly, I don't think you need to specify that the source is you, the authors, if this is your paper, in the caption!**

*The 3D numbers in the cube are darker now (Panel A, Figure 1), and their positions were changed in order to improve the visualization. The sigma lines have been changed in terms of thickness and color, being visually improved. Regarding Panel C, now there is a shadow area showing the respective standard deviation of both curves, enhancing the visualization of their differences. The word "source" was removed from the caption.*

[Figure]

**I also have an issue with the description in the text preceding the figure: you use the words "low to a high geometric diameter" and "broad to a narrow curve" but the differences in D and sigma are quite small: are they statistically significant? They are both pretty squarely in what we consider "accumulation" mode, so it would be helpful to see how much they overlap. How many PNSDs per season, are the numbers comparable? A bit more statistic could be useful here. In line 164 the authors say "broad statistical basis" but unclear 1) what they mean and 2) what the numbers actually are.**

*The changes mentioned in the geometric diameter and the sigma might seem small, but they can significantly change the overall number concentration. When we mentioned "low to a high geometric diameter" and "broad to a narrow curve" we were talking about variations in $D_g$ and sigma comparing the dry to the wet season of the accumulation mode (Figure 1c). The values expressed in Figure 1c for the dry season are N=1341 cm$^{-3}$, $D_g$=157 nm, and sigma=1.50 nm, and for the wet season N=301 cm$^{-3}$, $D_g$=164 nm, and sigma=1.39 nm. The difference is of about 7.4% in sigma and 4.3% in size, considering the highest values as the reference. For instance, the isolated effect of this change in sigma could reduce the difference between the two seasons in the number concentration by 9.7%. For the variation in $D_g$, the difference would increase by 5.0%. This means that smaller variations in size and sigma could account for significant changes in the number concentration.*

*In addition, these numbers are the output of in situ measurements, which means that they represent the intrinsic variations of the Amazonian aerosol population. Atmospheric models especially use these parameters to simulate aerosol concentration, where they often fix them. Therefore, we must report the variations we encounter since future modeling studies could use them as a reference or motivation for testing different values.*

*The number of valid fits per season in a 5-minute resolution is ~20800 in the wet season (February to May) and ~21500 in the dry season (October to January). These numbers were added to the text. We remember that our analyses only considered fits of $R^2$>0.9.*

*In order to improve the text and fit your comment, we changed the sentence you paraphrased to "smaller to a bigger…" and "broader to a narrower…", and included the sentence: The changes in sigma and $D_g$ are of about 7.4% and 4.3%, respectively, considering the highest values as the reference, which can account for significant changes in the total number concentration.*

**Line 184: "Dominated basically by N, the accumulation mode PNSDs are distributed in a geometric figure that resembles a cylinder" That sounds very qualitative, and I can't really see a cylinder (more a cone). Anyway, not sure what this phrase adds!**

*This part of the analysis is qualitative since we wanted to describe the arrangements. A similar approach has been used by McFarquhar et al. (2014) in describing the population of droplet size distribution, which they described as an ellipsoid. This qualitative analysis helps describe the most likely range of the fitted parameters and, therefore, the development of parameterizations. With a description of a volumetric figure, multiple relationships between the parameters could be used by models instead of fixed ones.*

**In general, Figure 2 could also do with a higher quality: better to have the three plots larger, and more defined. Very hard to tell anything. Potentially better to have the 2D projections as separate plots to appreciate them more.**

*The 2D plots are presented in the Supplement (Fig S2-6) and the values plotted are in Table S1.*

**Line 205: "initial" state.**

*It has been changed as recommended.*

**Line 226: I'm sure the authors can find a more professional term than "birth" here!**

*The referred word has been changed. The new sentence is: "Therefore, the core seen can be related to distributions capturing the  arising of these particles."*

**Figure 3: absolutely impossible to determine where the seasons actually are. No point in having the 3D version here, as it's not informative. Suggest changing to 9 plots (3x3) showing the three projections for the three modes, with seasons explicitly written there. Same for Fig. 4, except with 4 points this is easier to follow**

*Each plot has a reference marker to help the visualization. For the seasonal and the diurnal plot, December/January (DJ) and the 0 LST have the cross-maker, respectively, including in the projections. Following the months or the hours using the cross-marker as a reference helps the visualization. However, as discussed above, we are including the 2D plots in the Supplement and a table with all the actual values, so we hope this problem you are pointing out is diminished.*

**Line 246: "confirm" this cycle (you often use the present tense to talk about your work elsewhere, then switch to the past. Use some consistency!).**

*It has been changed as recommended.*

**Line 259: "happening"**

*It has been changed as recommended.*

**Line 315: "Many studies have been reporting" -> "have reported"**

*It has been changed as recommended.*

**Line 393: "The cause of this pattern showed to be linked to the diurnal cycle of precipitation, possibly associated with new particle formation" this should be rewritten for clarity! In general, the conclusions are of a slightly lower quality (clarity-wise) than other pieces of the manuscript. Not a big deal, but it would be good to improve this a bit more!**

*We rephrased the sentence as follows: These occurrences of the sub-20 nm fits in the morning and afternoon could be linked to the occurrence of precipitation since, in Central Amazon, the precipitation follows the same pattern and is known to be associated with new particle formation.*

---

## Author Comment (AC2)

**Response to Anonymous Referee**

**Article: "Amazonian Aerosol Size Distributions in a Lognormal Phase Space: Characteristics and Trajectories", by Gabriela R. Unfer et al., Egusphere 2023-1361.**

Dear Editor,

The authors thank both reviewers for their helpful comments and suggestions. Our responses to each comment are developed hereafter, along with an indication of changes made in the revised version of the text. As a summary, the revisions to the manuscript include the following highlights:

- All plots of the Results section have a version in 2D as scatter plots of N versus $D_g$ with sigma in a color scale presented in the Supplement.

- A table containing the summary of all fit parameters is presented in the Supplement.

- A more detailed description of the code-fitting methodology and a discussion about the separated analysis of the modes sub-50 nm and 50-100 nm (Aitken).

- The text was improved in readability and the abstract was rewritten.

- New references were added to improve the discussions.

The individual reviewer comments and responses are included in the following document, where reviewer comments are presented in **bold** and the author comments in *italics*.

Sincerely,

Gabriela R. Unfer, on behalf of all co-authors

**Reviewer 2:**

**Main comments:**

**This paper presents analysis of long-term aerosol particle number size distribution (PNSD) data from the ATTO tower through characterising the PNSD in a three-dimensional phase space represented by 1) the geometric mean diameter; 2) the geometric standard deviation; 3) the number concentration of lognormal modes fitted to the PNSD. The manuscript deals with an important topic: finding new ways to analyse the emerging long-term data sets of aerosol particle characteristics, which are critical for enhancing our understanding on e.g. aerosol-cloud-climate interactions, and is therefore approapriate for the scope of ACP. I think the approach for investigating trends and behavior of the PNSD as a function of various environmental parameters is interesting and novel, although the exact value for potential applications is to be demonstrated in future studies. The manuscript therefore has scientific value that is of potential interest for the readership of ACP. There are, however, some aspects of the manuscript that should be improved before publication can be recommended - related to both the scientific approach and the presentation quality:**

*Dear Reviewer, we would like to thank you for your comments and suggestions; they were significant in improving and clarifying some essential aspects of the manuscript content. In the Editor's letter, we explain the main changes in the manuscript, and below, we listed these aspects related to your recommendations.*

**General:**

**- The reference list of the manuscript is rather limited with a high proportion of work from the authors themselves. I understand that when it comes to PNSDs in the tropical atmosphere this is hard to avoid, but perhaps there is previous work on PNSD characteristics from other parts of the world worth mentioning here? Please consider also adding some discussion and comparisons to past work to the Discussion and conclusions section.**

*As recommended, we increased the diversity of the cited works and included comparison discussions on the text. Please find below the new references added.*

Boucher, O: Atmospheric Aerosols: Properties and Climate Impacts, 1, Springer Dordrecht, https://doi.org/10.1007/978-94-017-9649-1, 2015.

Dada, L., Paasonen, P., Nieminen, T., Buenrostro Mazon, S., Kontkanen, J., Peräkylä, O., Lehtipalo, K., Hussein, T., Petäjä, T., Kerminen, V.-M., Bäck, J., and Kulmala, M.: Long-term analysis of clear-sky new particle formation events and nonevents in Hyytiälä, Atmos. Chem. Phys., 17, 6227–6241, https://doi.org/10.5194/acp-17-6227-2017, 2017.

Khadir, T., Riipinen, I., Talvinen, S., Heslin-Rees, D., Pöhlker, C., Rizzo, L., Machado, L. A. T., Franco, M. A., Kremper, L. A., Artaxo, P., Petäjä, T., Kulmala, M., Tunved, P., Ekman, A. M. L., Krejci, R., and Virtanen, A.: Sink, Source or Something In-Between? Net Effects of Precipitation on Aerosol Particle Populations, Geophys. Res. Lett., 50(19). https://doi.org/10.1029/2023GL104325, 2023.

Kulmala, M., Petäjä, T., Nieminen, T., Sipilä, M, Manninen, H. E., Lehtipalo, K., Dal Maso, M., Aalto, P. P., Junninen, H., Paasonen, P., Riipinen, I., Lehtinen, K., Laasonen, A., and Kerminen, V-M.: Measurement of the nucleation of atmospheric aerosol particles. Nature Protocols, 7, 1651–1667, https://doi.org/10.1038/nprot.2012.091, 2012.

Mäkelä, J. M., Koponen, I. K., Aalto, P., Kulmala. M.: One-year data of submicron size modes of tropospheric background aerosol in Southern Finland, Journal of Aerosol Science, 31, 595-611, https://doi.org/10.1016/S0021-8502(99)00545-5, 2000.

Sogacheva, L., Saukkonen, L., Nilsson, E. D., Dal Maso, M., Schultz, D. M., De Leeuw, G., and Kulmala, M.: New aerosol particle formation in different synoptic situations at Hyytiälä, Southern Finland, Tellus B: Chemical and Physical Meteorology, 60:4, 485-494, https://doi.org/10.1111/j.1600-0889.2008.00364.x, 2008.

Tunved, P., Hansson, H.-C., Kulmala, M., Aalto, P., Viisanen, Y., Karlsson, H., Kristensson, A., Swietlicki, E., Dal Maso, M., Ström, J., and Komppula, M.: One year boundary layer aerosol size distribution data from five nordic background stations, Atmos. Chem. Phys., 3, 2183–2205, https://doi.org/10.5194/acp-3-2183-2003, 2003.

Wang, J., Krejci, R., Giangrande, S., Kuang, C., Barbosa, H. M. J., Brito, J., Carbone, S., Chi, X., Comstock, J., Ditas, F., Lavric, J., Manninen, H. E., Mei, F., Moran-Zuloaga, D., Pöhlker, C., Pöhlker, M. L., Saturno, J., Schmid, B., Souza, R. A. F., Springston, S. R., Tomlinson, J. M., Toto, T., Walter, D., Wimmer, D., Smith, J. N., Kulmala, M., Machado, L. A. T., Artaxo, P., Andreae, M. O., Petäjä, T., and Martin, S. T.: Amazon boundary layer aerosol concentration sustained by vertical transport during rainfall, Nature, 539(7629), 416–419. https://doi.org/10.1038/nature19819, 2016.

**- The manuscript needs to be improved for readability and precise use of terms and language. Some examples are provided in the specific comments below, but generally e.g. references to analysis or figures that are only to be presented later in the manuscript should be kept to minimum, and concepts and terms should be referred to as precisely as possible.**

*Based on your suggestions, the full text was improved for readability, and the mention of later analysis/figures was minimized.*

**- It would be interesting if the authors could, for example at the end of the introduction section, reflect on the potential research questions that could be answered using the**

**approach presented here - beyond following temporal patterns and responses to precipitation events. What larger-scale implications might the results presented here have?**

*We envision that the analysis of aerosol population in the lognormal phase space can be helpful, for example, in understanding the distribution of particles under different synoptic systems or even interannual variabilities, like in El Nino/La Nina. It is also possible to compare different global warming scenarios regarding aerosol distributions. One could analyze the phase space for different geographic regions by plotting the different aerosol populations and analyzing how they cluster in the phase space. Another possibility is plotting particle growth for different new particle formation events and possibly extracting the parameterizations.*

*This discussion has been added to the introduction section*

**- My main potentially scientifically major comment has to do with the fact that it appears from Fig. 2 and the text that the division between the "sub-50 nm" and "Aitken" modes was somewhat arbitrary - if I understand correctly, a predefined size-cut at 50 nm was simply used instead of letting the fitting algorithm find the best number of modes and their parameters. This constraint makes it difficult to use the fitted "sub-50" and "Aitken" mode bevior for analysis of the underlying microphysics and chemistry - because it is not clear whether one can justify the choice of the modes in terms of them representing clearly different aerosol populations. Could the authors please clarify their methodological choices in this regard and reflect on the potential implications for the interpretation and usefulness of the presented results?**

*In Central Amazon, where our measurements were taken, the maximum number of modes is three, as presented and discussed in Franco et al. (2022). In our study, three modes were not necessarily always fitted; the code was free to decide between one and three. Regarding the diameter ranges, the study of Franco et al. (2022) and other studies like Machado et al. (2021) have already shown that in the ATTO region, the size ranges of 10 to 50 nm, 50 to 100 nm, and 100 to 400 nm are representatives of the aerosol modes and that they present different behaviors.*

*Specifically, the separation of the literature Aitken mode into two (sub-50 nm and 50 to 100 nm) did not affect the behavior of our Aitken mode itself since the concentration of the sub-50 nm mode is low compared to the other modes. Furthermore, it was beneficial since we could see clearly from the analyses that the sub-50 nm mode presented distinct results from the Aitken mode. Machado et al. (2021) already had shown that this mode presents an increase in concentration after rainfall events, in contrast with the Aitken (50-100 nm) and accumulation modes, which present a decrease in concentration. The segregation of the Aitken mode into two brings rich information about the formation of new particles in the Amazon.*

*We improved the discussion of the modes fitting and modes selections in the methodology section to clarify the lognormal fit. In addition, the typical dry and wet season distributions are in the Supplement (Fig. S1), which corroborates our methodology.*

**Specific comments:**

- Abstract, p. 1, e.g. lines 23-24 and 25-26: The abstract should be stand-alone and understandable without having to read the manuscript in detail. It is very difficult to understand e.g. what it means that "the sub-50 nm mode appears as a curved cone, the Aitken mode as a semi-sphere, and the accumulation mode as a cylinder" without looking at the plots in the manuscript. Also what does a "positive linear slope" mean in "The diurnal cycle of sub-50 nm particles in the dry season shows a positive linear slope as a function of all three fit parameters." - i.e. which variable has a positive linear slope as a function of the three fit parameters? Do you perhaps mean that all three fit parameters have a positive linear slope as a function of time? Please revise the abstract for readability through e.g. defining which modes where fitted to the data, accurate definition of parameters and clearly highlighting the key conclusions that can be summarized without having to read the entire mansucript.

*The abstract was rewritten and improved. Regarding the positive linear slope, we meant the change of every parameter with respect to time. But for clarity, we changed it to just "a linear cycle", since it depends on the starting point of the time to determine whether it is positive or negative, but the trajectory is linear overall.*

- p. 2, lines 46-48: PLease revise the sentence starting with "Improving aerosols parameterizations..." for clarity, English language and readability.

*The sentence was rephrased.*

- p. 2, line 64 and p. 3, line 66: What do you refer to with the term "comparatively" - as compared with what? Please specify if possible.

*The term refers to the comparison between the concentration in each of the seasons to the typical concentration in the Amazon and also worldwide. This clarified sentence was added to the text.*

- p 3, lines 89-91: Perhaps it is appropriate also to refer to the study by Hussein et al. from 2005 (Hussein, T., Dal Maso, M., Petäjä, T., Koponen, I. K., Paatero, P., Aalto, P. P., Hämeri, K., & Kulmala, M. (2005). Evaluation of an automatic algorithm for fitting the particle number size distributions. Boreal Environment Research, 10(5), 337-355. http://www.borenv.net/BER/pdfs/ber10/ber10-337.pdf) here? Also, what do you mean by "isolating the variability of isolated modes" - please consider revising for readability.

*The reference was included and the referred sentence was improved. We meant that when working with the lognormal fit, one can show the variability of every mode separately.*

**- p. 4, lines 130-131: Was it always appropriate to fit three modes or were there instances when a different number of modes would have represented the size distribution better? If yes, what kind of error might the choice of three modes introduce to the results presented? Furthermore, did you fix the size ranges assumed for the three modes or let the code decide this. What might this imply for the results? Please add a brief elaboration on these questions and a justification of the chosen approach (in terms of numbers of modes and size ranges assumed).**

*The code did not necessarily fit three modes. It was free to fit between one and three. In addition, the decision of three modes and the range in diameter was based on a statistical analysis of 6 years of data (Figure 2 in Franco et al., 2022), where the maximum of modes and the ranges is clear.*

*The mode fitting code started with a fixed guess based on the statistical diameter position of each one of the three modes. Later, the code did two optimizations to correct the guess: one was done based on the other modes' positions, and the other was an optimization of the three modes together. After these three steps, it is expected to obtain more precise fits. We studied the modes separately in our analyses, considering all the fitted distributions in 1 year. So, in the end, we had enough data points for every mode, but not necessarily they were always fitted together in the time resolution of 5 minutes. However, since we present the means in our analyses, we have the statistical representation of every mode for each studied case.*

**- p. 7, line 72: Please revise the expression commenting that "the SMPS used was limited to 10 nm" for accuracy. I guess you want to say that the lower detection limit of the SMPS was 10 nm.**

*The referred expression was improved as recommended.*

**- p. 7, line 76: Instead of "constant dispersion" do you mean "constant standard deviation"? Can this depend on the the size ranges that you have constrained (if that is the case)?**

*Yes, by constant dispersion we mean constant standard deviation, as seen by the projection (shadow) on the sigma axis. Regarding the size ranges, the code first considered the same range of sigma for all modes, varying from 1.1 to 1.55 nm, and then in the optimization step, the code allowed a new range of the maximum of 1.2 times the first fit.*

*Although the ranges in the axis are different, in Fig. 2, it is noticeable that the variations in the sigma values of the sub-50 nm and the accumulation are more spread out vertically than the ones in the Aitken mode, which is nearly around 1.2 and 1.3 nm. You can check this on the new 2D plot in the Supplement. In Fig. S2b the colors are mainly green, exactly between 1.2 and 1.3 nm, while the others (Figures S2a and S2c) have a greater variation.*

**- p. 8, lines 215-216: Please revise the sentence starting "the accumulation mode dominates...". I believe you want to say that "accumulation mode dominates over Aitken mode" and at the end "prevalent" instead of "equivalent".**

*You are correct when we tried to say that the accumulation mode dominates over the Aitken mode. For the second part, the word "equivalent" fits better since we meant that both modes have the same overall concentration.*

**- p. 9, lines 242-243, the sentence saying "reaching a maximum during the night probably due to late afternoon rainfalls": What about the importance of boundary layer dynamics?**

*We included in the text the effects of the nocturnal boundary layer in the sub-50nm concentration dynamics. Rainfall increases the ultrafine particle number, and the nocturnal boundary layer keeps the concentration nearly constant during the night.*

**- p. 9, line 243: with "initiates" do you perhaps mean "begins"?**

*Exactly.*

**- p. 12, lines 305-306: Please revise the the sentence "Since lightning and precipitation peak simultaneously (Mattos et al., 2017) the following results (Figure 5) are intercomparable, promoting a complete characterization of the aerosol-precipitation interaction." for clarity and readability.**

*Thank you for the comment. The text has been improved and we hope it is clear now.*

**- p. 13, line 314: Instead of "sensible" do you perhaps mean "pronounced" or something similar?**

*Exactly. The word has been changed as recommended.*

**- p. 13, line 317: Please revise the sentence starting as "In addition, it was shown..." for readability and English language.**

*The sentence has been rephrased and we hope it is now improved.*

**- Same as above, the sentence starting "In fact, it can be seen...": Where exactly can this be seen? Please specify.**

*The whole paragraph has been rewritten and we hope it is improved now.*

**- p. 14, line 327: Please revise the sentence for readability, clarity and precise use of English. Specifically, what do you mean by "background trajectories" - please specify.**

*The text has been improved and we hope it is clear now. The new sentences are: "The following analysis explored the background aerosol concentration in the morning considering afternoons with and without precipitation. It was considered time trajectories from 6 to 12 LST."*